# Partially Hydrolyzed Guar Gum Suppresses the Development of Sarcopenic Obesity

**DOI:** 10.3390/nu14061157

**Published:** 2022-03-09

**Authors:** Takuro Okamura, Masahide Hamaguchi, Jun Mori, Mihoko Yamaguchi, Katsura Mizushima, Aya Abe, Makoto Ozeki, Ryoichi Sasano, Yuji Naito, Michiaki Fukui

**Affiliations:** 1Department of Endocrinology and Metabolism, Graduate School of Medical Science, Kyoto Prefectural University of Medicine, Kyoto 602-8566, Japan; d04sm012@koto.kpu-m.ac.jp (T.O.); mhama@koto.kpu-m.ac.jp (M.H.); 2Department of Pediatrics, Graduate School of Medical Science, Kyoto Prefectural University of Medicine, Kyoto 602-8566, Japan; jun1113@koto.kpu-m.ac.jp (J.M.); m-ymgch@koto.kpu-m.ac.jp (M.Y.); 3Department of Human Immunology and Nutrition Science, Kyoto Prefectural University of Medicine, Kyoto 602-8566, Japan; mizushima@koto.kpu-m.ac.jp (K.M.); ynaito@koto.kpu-m.ac.jp (Y.N.); 4Nutrition Division, Taiyo Kagaku Co., Ltd., Yokkaichi 510-0844, Japan; aabe@taiyokagaku.co.jp (A.A.); mozeki@taiyokagaku.co.jp (M.O.); 5AiSTI Science Co., Ltd., Wakayama 640-8390, Japan; sasano@aisti.co.jp

**Keywords:** partially hydrolyzed guar gum, PHGG, sarcopenic obesity, metabolite, innate immunity

## Abstract

Partially hydrolyzed guar gum (PHGG) is a soluble dietary fiber derived through controlled enzymatic hydrolysis of guar gum, a highly viscous galactomannan derived from the seeds of *Cyamopsis tetragonoloba*. Here, we examined the therapeutic potential of dietary supplementation with PHGG against sarcopenic obesity using Db/Db mice. Db/Db mice fed a normal diet alone or a fiber-free diet, or supplemented with a diet containing PHGG (5%), were examined. PHGG increased grip strength and the weight of skeletal muscles. PHGG increased the short-chain fatty acids (SCFAs) concentration in feces and sera. Concerning innate immunity, PHGG decreased the ratio of inflammatory cells, while increasing the ratio of anti-inflammatory cells in the small intestine. The present study demonstrated the preventive effect of PHGG on sarcopenic obesity. Changes in nutrient absorption might be involved through the promotion of an anti-inflammatory shift of innate immunity in the intestine accompanied by an increase in SCFA production by PHGG.

## 1. Introduction

Recently, muscle atrophy has been considered as a complication of diabetes mellitus [1]. It has been found that muscle atrophy, i.e., sarcopenia and sarcopenic obesity, is strongly correlated with dietary patterns and metabolic abnormalities [2,3]. We have previously demonstrated the presence of muscle atrophy in diabetic patients [4,5]. Muscle atrophy has also been reported to be a risk factor for decreased activities of daily living and mortality [6,7]; in particular, sarcopenic obesity has been reported to accelerate the loss of muscle mass and function, reduce physical performance, and increase the risk of death [8], but no effective treatment has been found.

There have been various reports on the dietary treatment of sarcopenic obesity; however, the effectiveness of calorie restriction has not been reported [8], while protein [9], calcium, and Vitamin D supplements [10] have been shown to have some efficacy in sarcopenic obesity. Optimizing protein intake while restricting calories requires careful medical monitoring and diet planning, often under the auspices of a registered dietitian with expertise in this population.

On the other hand, the efficacy of digestible oligosaccharides [11,12], plant products [13], lactobacilli [14], and polysaccharides [15,16] against sarcopenia and obesity has been reported. Partially hydrolyzed guar gum (PHGG) is a soluble dietary fiber derived through controlled enzymatic hydrolysis of guar gum, a highly viscous galactomannan derived from the seeds of *Cyamopsis tetragonoloba*. PHGG has not only been reported to improve the symptoms of irritable bowel syndrome (both constipation and diarrhea type) [17], but it has also been reported to be useful in the treatment of metabolic syndrome such as abnormal glucose and lipid metabolism and non-alcoholic fatty liver disease [18,19,20]. PHGG is not digested in the upper digestive tract, but is fermented by gut microbiota in the lower digestive tract, producing large amounts of short-chain fatty acids (SCFAs). PHGG also promotes an increase in beneficial bacteria such as Bifidobacterium and butyrate-producing bacteria in the intestine [21]. In fecal culture, PHGG promotes the production of not only butyric acid but also acetic acid and propionic acid [22]. Thus, PHGG alters the composition of the microbiota, and increases metabolites such as butyric acid, acetic acid, and various amino acids [23,24,25]; therefore, PHGG could have beneficial health effects on the host through changes in the gut microbiota and its metabolites. Furthermore, dysbiosis has been reported to contribute to intestinal barrier dysfunction, which facilitates the transfer of lipopolysaccharide and endotoxin into the circulation, leading to systemic chronic inflammation and insulin resistance, and eventually to sarcopenia [26]. Since there are many reports on the relationship between obesity and dysbiosis, it is expected that there is also a relationship between sarcopenia obesity and altered gut microbiota. In addition, PHGG supports the effects of voluntary exercise on obesity and glucose intolerance due to a high-fat diet [27]. For the above reasons, we hypothesized that PHGG might be an effective new treatment for sarcopenia obesity, which has not been clarified. The present study was designed to examine the influence of PHGG on sarcopenic obesity and to investigate the related hypothetical mechanism.

## 2. Materials and Methods

### 2.1. Mice

All experimental procedures were approved by the Committee for Animal Research, Kyoto Prefectural University of Medicine (M2020-41). Seven-week-old male diabetic homozygous Db/Db mice were purchased from Shimizu Laboratory Supplies (Kyoto, Japan). The mice were eight weeks old at the beginning of the experimental procedures. For eight weeks, they were fed either a normal diet (ND; AIN-93G, 361.2 kcal/100 g, fat kcal 7.0% soybean oil, cellulose 5%/*w*; CLEA Japan, Tokyo, Japan), a fiber-free diet (FFD; Modified AIN93G rodent diet in which cellulose was replaced with corn starch (5%/*w*), 388.1 kcal/100 g), or a modified AIN93G rodent diet in which cellulose was replaced with commercially available, partially hydrolyzed guar gum (PHGG; Modified AIN93G rodent diet without cellulose added PHGG (5%/*w*), 378.2 kcal/100 g) (Sunfiber^®^, Taiyo Kagaku Co., Ltd., Yokkaichi, Japan) (Table 1). The PHGG used in this study was provided by Taiyo Kagaku (Yokkaichi, Japan), and we asked the Oriental Bio Service (Kyoto, Japan) to mix it with AIN-93G to create a solid food. Six mice were divided into the following three groups: (1) Db/Db with ND (ND), (2) Db/Db with FFD (FFD), and (3) Db/Db with PHGG (PHGG) (Figure 1A). Paired feeding was performed by supplying the same amount of feed. The mice were maintained in an environmentally controlled room (temperature, 23 ± 1.5 °C; humidity, 40–60%) with a 12 h light/12 h dark cycle (lights on at 7 am and off at 7 pm). Cages with a size of W220 × L320 × H135 mm were used, and three mice were kept in each cage. The cumulative oral intake was measured for 8 weeks. The manually weighed fresh food was placed in a trough in each cage once every three days in the morning (9 a.m.), and after 24 h, the amounts of food were measured. The remnants of chow were discarded. At 16 weeks of age, an overnight fast was administered and all the mice were sacrificed by the administration of a combination anesthetic: 4.0 mg/kg, midazolam, 0.3 mg/kg of medetomidine, and 5.0 mg/kg of butorphanol [28]. Since ketamine was designated as a narcotic in 2007 in Japan, the combination anesthetic has been used in our laboratory.

### 2.2. Analytical Procedures for the Glucose and Insulin Tolerance Tests

To measure weight, the mice were fasted for 14 h overnight and their weight was measured once a week. In 15-week-old mice, an intraperitoneal glucose tolerance test (iPGTT) (2 g/kg body weight) and an insulin tolerance test (ITT) (0.5 U/kg body weight) were performed after 14 h and 5 h fasts, respectively, and blood glucose was measured with blood collected by orbital puncture and as indicated by a glucometer (Gultest Neo Alpha; Sanwa Kagaku Kenkyusho, Nagoya, Japan). The area under the curve (AUC) of the iPGTT and ITT results was analyzed [29].

### 2.3. Indirect Calorimetry

In vivo indirect open-circuit calorimetry was conducted at a controlled ambient temperature (24 ± 2 °C) in metabolic cages. A constant airflow (0.6 L/min) was channeled into the cage and measured using a metabolic analyzer (O_2_/CO_2_ Analyzer MM202R; Muromachi Kikai Co., Ltd., Tokyo, Japan). VO_2_ and VCO_2_ were measured for 48 h during 12 h light/12 h dark cycles. At the inlets and outlets of the sealed cages, concentrations of gas were measured. During the experiments, the mice were allowed to drink water freely and food was controlled according to experimental methods. The O_2_/CO_2_ analyzer measured the VO_2_ and VCO_2_ in the cage every 3 min and 20 pieces of data per hour were obtained. In each group of six animals, the mean and SD values of the light and dark periods were calculated.

### 2.4. Measurement of Grip Strength

Grip strength was assessed by a strength meter for mice (model DS2-50N, IMADA Co., Ltd., Toyoashi, Japan) in the 16-week-old mice. Measurements were taken six consecutive times daily at 1 min intervals, and investigators were blinded to experimental conditions. Grip strength was standardized by body weight [30].

### 2.5. Muscle Histology

The soleus and plantaris muscles were fixed with 10% buffered formaldehyde and embedded in paraffin. Muscle sections were prepared and stained with hematoxylin and eosin. Images were captured with a BZ-X710 fluorescence microscope (Keyence, Osaka, Japan), and the cross-sectional area was measured using ImageJ software version 1.53k (NIH, Bethesda, MD, USA).

### 2.6. Liver Histology

The liver was fixed with 10% buffered formaldehyde and embedded in paraffin. Its sections were prepared and stained with hematoxylin and eosin and Masson’s trichrome stain. Images were captured using a BZ-X710 fluorescence microscope (Keyence, Osaka, Japan). Moreover, we adopted the nonalcoholic fatty liver disease (NAFLD) activity score (NAS) to assess the severity of NAFLD [31], a well-known standard used for the assessment of nonalcoholic steatohepatitis (NASH) severity and changes in NAFLD. A trained hematopathologist, blinded to the experimental conditions, examined all the sections for steatosis, lobular inflammation, hepatocyte ballooning, and fibrosis, according to the NAS [31]. The scoring system comprised 14 histological features with four evaluated semi-quantitatively: steatosis (0–3), lobular inflammation (0–2), hepatocellular ballooning (0–2), and fibrosis (0–4). Moreover, to assess fibrosis, we used none (0); mild, zone 3, perisinusoidal (1A); moderate, zone 3, perisinusoidal (1B); portal/periportal (1C); perisinusoidal and portal/periportal (2); bridging fibrosis (3); and cirrhosis (4).

### 2.7. Biochemistry

Blood samples were taken from fasted mice and serum samples were collected after centrifugation at 14,000 rpm for 10 min at 4 °C. The level of alanine aminotransferase (ALT) was measured by a standardization support method of the Japanese Society for Clinical Chemistry [32], and total cholesterol (T-Chol) [33], triglycerides (TG) [34], and non-esterified fatty acids (NEFAs) [35] were measured by the enzymatic method. Biochemical examinations were performed using FUJIFILM Wako Pure 18 Chemical Corporation (Osaka, Japan).

### 2.8. Measurement of Free Fatty Acids in the Sera, Liver, Feces, and Plantaris Muscles

Gas chromatography–mass spectrometry (GC/MS), using an Agilent 7890B/7000D (Agilent Technologies, Santa Clara, CA, USA), was performed for measurement of the concentration of palmitic acid in murine sera, liver, feces, and plantaris muscle. Fecal pellets were collected by resection of the distal colon during sacrifice. A total of 25 milliliters of sera and 15 µg of the liver, feces, and plantaris muscle were methylated using a fatty acid methylation kit (Nacalai Tesque, Kyoto, Japan). The final product was loaded onto a Varian capillary column (DB-FATWAX UI; Agilent Technologies). For fatty acid separation, the capillary column used was CP-Sil 88 for FAME (length, 100 mm; inner diameter, 0.25 mm; membrane thickness, 0.20 μm; Agilent Technologies). The column temperature was held at 100 °C for 4 min, gradually increased to 240 °C at 3 °C/min, and maintained for 7 min. Samples were injected in split mode at a ratio of 5:1. Palmitic acid methyl ester was detected in the selective ion monitoring mode. All results were normalized to the peak height of the C17:0 internal standard [36].

### 2.9. Measurement of SCFA Levels in the Feces and Sera and Amino Acid Concentrations in the Plantaris Muscle, Feces, Sera, and Liver Samples

The Agilent 7890B/7000D GC/MS System (Agilent Technologies, Santa Clara, CA, USA) was used to analyze the SCFA composition of murine rectal feces and serum samples and the amino acid composition of murine plantaris muscle. First, rectal feces (20 mg), plantaris muscle (20 mg), serum (50 µL), and liver (20 mg) samples were added to 500 µL diluted water and 500 µL acetonitrile, and ground at 4000 rpm for 2 min in a ball mill. Second, the samples were then placed in a shaker for 30 min at 1000 rpm at 37 °C and centrifuged for 3 min at 14,000 rpm at room temperature. Third, the upper layer (500 μL) was removed, mixed with 500 μL acetonitrile, and placed in a shaker for 3 min at 1000 rpm at 37 °C. Finally, the samples were centrifuged for 3 min at room temperature at 14,000 rpm, and SCFAs and amino acids were extracted by adding 0.1 mol/L NaOH to adjust the pH to 8.

GC/MS system with the on-line solid-phase extraction (SPE) method was used for automatic measurement of SCFA and amino acid concentrations. The SPE–GC system SGI-M100 (IST Science, Wakayama, Japan) was designed to automatically inject the sample into the SPE and GC/MS after the sample was placed in the vial, and was placed on the autosampler tray. Solids were stratified by Flash-SPE ACXs (AiSTI Science, Wakayama, Japan). Aliquots totaling 50 microliters each of the sample extracts were collected, then loaded into the solid phase and washed with acetonitrile and water (1:1). Next, the samples were dehydrated with acetone, impregnated with 4 μL of N-tert-butyldimethylsilyl-N-methyltrifluoroacetamide (MTBSTFA)-toluene solution (1:3), derivatized in the solid phase and eluted with hexane. The final product was injected using a programmed temperature vaporizer (PTV) injector, LVI-S250 (AiSTI Science), maintained at 150 °C for 0.5 min, gradually increasing from 25 °C/min to 290 °C and maintained for 16 min. Samples were loaded onto a capillary column Vf-5ms (30 m × 0.25 mm (i.d.) × 0.25 μm (film thickness); Agilent Technologies). The column temperature was maintained for 3 min at 60 °C, then gradually raised to 100 °C at 10 °C/min and 310 °C at 20 °C/min, and maintained at 310 °C for 7 min. Samples were instilled in a split mode at a ratio of 20:1. Each SCFA was found in scan mode (*m*/*z*: 70–470). All values were standardized to a peak height of tetradeuteroacetic acid (0.02 nmol/μL) for SCFA and noreucine (0.02 nmol/μL) for amino acids [29].

### 2.10. Gene Expression Analysis in Murine Liver, Jejunum, White Adipose Tissue, and Soleus Muscle

Liver, jejunum, white adipose tissue, and soleus muscle from mice fasted for 16 h were excised and quickly cryopreserved in liquid nitrogen. Liver samples were then homogenized in ice-cold QIAzol Lysis reagent (Qiagen, Venlo, The Netherlands), and total RNA was isolated as described in the manufacturer’s instructions. Total RNA (0.5 µg) was subjected to first-strand cDNA synthesis using the High-Capacity cDNA Reverse Transcription Kit (Applied Biosystems, Foster City, CA, USA) with oligonucleotide dT primers; reverse transcription was performed according to the manufacturer’s recommendations by random hexamer priming. Reverse transcription reactions were performed at 37 °C for 120 min and reverse transcription was performed at 85 °C for 5 min. Real-time reverse transcription–polymerase chain reaction (RT-PCR) was used to quantitate the mRNA expression levels of *Tnfa*, *Il6*, *Il1b*, *Col1a*, and *Il22* in the liver; *Tnfa*, *Il6*, and *Col1a* in the white adipose tissue; *Trim63*, *Fbxo32*, *Tnfa*, *Il6*, *Foxo1*, and *Hdac4* in the soleus plantaris muscle; and *Tnfa*, *Il6*, *Il1b*, *Cd36*, *Slc15a*, and *Il22* in the jejunum. TaqMan Fast Advanced Master Mix (Applied Biosystems) was used for RT-PCR according to the manufacturer’s instructions. The following PCR conditions were used: one cycle at 50 °C for 2 min and at 95 °C for 20 s, followed by 40 cycles at 95 °C for 1 s and at 60 °C for 20 s.

The relative expression levels of each targeted gene were normalized to *G**apdh* threshold cycle (Ct) values and quantified using the comparative threshold cycle 2^−ΔΔCT^ method as previously described [29]. Signals from ND mice were set to a relative value of 1.0. Six mice from each group were examined, and RT-PCR was conducted for each sample in triplicate.

### 2.11. Small Intestine Histology

Tissue from the small intestine was fixed with 10% buffered formaldehyde and embedded in paraffin. Tissue sections were prepared and stained with hematoxylin and eosin. Images were captured using a fluorescence microscope (BZ-X710). Villus height/width and crypt depth were analyzed using ImageJ software.

### 2.12. Protocol for Isolation of Mononuclear Cells from Small Intestines of Mice

To eliminate blood contamination of the small intestine, systemic perfusion with heparinized saline was performed prior to harvesting its tissue or washing with phosphate-buffered saline (PBS). The Lamina Propria Dissociation Kit (130-097-410; Miltenyi Biotec, Bergisch Gladbach, Germany) was used for the isolation of intestinal lamina propria (LPL) mononuclear cells according to the instructions. Cell pellets were resuspended in 40% Percoll^®^ and slowly added to the top of a centrifuge tube containing 5 mL of 60% Percoll^®^ at the bottom; washed twice with 2% FBS/PBS and centrifuged (420× *g*, 20 min) after density gradient to obtain LPL mononuclear cells.

### 2.13. Tissue Preparation and Flow Cytometry

Stained cells were analyzed using FACS Canto II, and the data were analyzed using FlowJo version 10 software (TreeStar, Ashland, OR, USA). We used the following antibodies for the gating of innate lymphoid cells: Biotin-CD3e (100304; clone: 145-2C11; 1/200; eBioscience, San Diego, CA, USA), Biotin-CD45R/B220 (103204; clone: RA3–6B2; 1/200; eBioscience), Biotin-Gr-1 (108404; clone: RB6-8C5; 1/200; eBioscience), Biotin-CD11c (117304; clone: N418; 1/200; eBioscience), Biotin-CD11b (101204; clone: M1/70; 1/200; eBioscience), Biotin-Ter119 (116204; clone: TER-119; 1/200; eBioscience), Biotin-FceRIa (134304; clone: MAR-1; 1/200; eBioscience), FITC-Streptavidin (405202; 1/500; eBioscience), PE-Cy7-CD127 (135014; clone: A7R34; 1/100; eBioscience), Pacific Blue-CD45 (103116; clone: 30-F11; 1/100; eBioscience), PE -GATA-3 (clone TWAJ, 1/50; eBioscience), APC -RORγ (clone AFKJS-9, 1/50, eBioscience), and Fixable Viability Dye eFluor 780 (1/400; eBioscience) [37,38] (Appendix A). In addition, we used the following antibodies for gating of M1 and M2 macrophages: APC-CD45.2 (17045482; clone: 104, 1/50; eBioscience), PE-F4/80 (12480182; clone: BM8, 1/50, eBioscience), APC-Cy7-CD11b (47011282; clone: M1/70, 1/50; eBioscience), FITC-CD206 (MA516870; clone: MR5D3, 1/50, eBioscience), and PE-Cy7-CD11c (25011482; clone: N418, 1/50, eBioscience) [39] (Appendix A).

### 2.14. Statistical Analyses

The data were analyzed using JMP software ver. 13.0 (SAS, Cary, NC, USA). One-way analysis of variance was used to compare different groups. *p*-values < 0.05 were accepted as statistically significant. Figures were generated using the GraphPad Prism software (version 9.0; San Diego, CA, USA).

## 3. Results

### 3.1. Glucose Intolerance, Which Was Worsened by FFD, Was Significantly Improved by PHGG Administration

Bodyweight was heaviest in the PHGG group, followed by the FFD and ND group (Figure 1B). Dietary intake was equivalent among the groups according to paired feeding (Figure 1C). In addition, glucose tolerance (Figure 1D,E) and insulin sensitivity (Figure 1F,G) in the FFD group significantly reduced, compared to those in the ND group, whereas those in the PHGG group were significantly improved, compared to those in the ND or FFD group.

### 3.2. FFD Decreased the Energy Metabolism, Whereas PHGG Increased

Next, energy metabolisms were measured by housing 16 weeks of age mice in a metabolic cage. O_2_ consumption and CO_2_ content, which indicate energy expenditure, were significantly decreased in the FFD group, compared to the ND group; on the other hand, the energy metabolism increased in the PHGG group (Figure 1H–K).

### 3.3. PHGG Decreased Hepatic Enzyme and Improved Lipid Metabolism

In the biochemistry test, levels of ALT and T-chol in the PHGG group significantly decreased, compared with those in the ND and FFD groups, and the levels of TG and NEFA in the PHGG group significantly decreased, compared with those in the FFD group (Figure 1L–O).

### 3.4. PHGG Improved Intrahepatic Fat Accumulation and Fibrosis and Decreased the Gene Expression Related to Inflammation and Fibrosis in the Liver and White Adipose Tissue

Representative histological images of the liver are shown in Appendix A. NAFLD activity score in the FFD group was significantly higher than that in the ND group, whereas that in the PHGG group was significantly lower than that in the ND or FFD group (Appendix A). In addition, intrahepatic fat accumulation evaluated by oil red-O staining of the liver in the PHGG group significantly decreased, compared to that in the ND or FFD group (Appendix A).

In RT-PCR analyses, the expression of genes involved in the inflammatory response, such as Tnfa, Il6, and Il1b, and fibrosis, such as Col1a, in the liver of the FFD group was significantly higher than that of the ND group, whereas that of the PHGG group was significantly lower than that of the FFD group (Appendix A). On the other hand, the expression of the IL-22 gene, a cytokine which is downstream of ILC3 and reported to ameliorate liver damage in NAFLD [34,35,36], was significantly decreased in the FFD group compared to the ND group, whereas it was significantly increased in the PHGG group (Appendix A). Moreover, the expression of Tnfa, Il6, and Col1a in white adipose tissue of the FFD group was significantly higher than that of the ND and PHGG groups, whereas those expressions of the PHGG group were significantly lower than those of the ND and FFD groups (Appendix A).

In flow cytometric analyses, M1/M2 macrophages ratio in the liver of the FFD group was higher than that of the ND group, whereas that in the PHGG group was lower than that in the ND and FFD groups (Appendix A). The ratio of ILC1s in CD45-positive cells was not different between the three groups (Appendix A); on the other hand, the ratio of ILC3s in CD45-positive cells in the PHGG group was significantly lower than that in the ND and FFD groups (Appendix A) and the ratio of ex-ILC3s in CD45-positive cells in the PHGG group was significantly lower than that in the ND and FFD groups (Appendix A).

### 3.5. The Administration of PHGG Improved Sarcopenic Obesity

Absolute and relative grip strength in the FFD group was significantly lower than that in the ND group, and administration of PHGG significantly increased the absolute and relative grip strength (Figure 2A,B). Absolute epididymal fat weight was not different between the three groups; however, relative epididymal fat weight in the FFD was significantly higher than that in the ND group. The administration of PHGG significantly decreased the relative epididymal fat weight, compared to the ND and PHGG groups (Figure 2C and Figure 3D). Moreover, absolute and relative cecum weight in the FFD group was significantly lower than that in the ND group, whereas that in the PHGG group was significantly higher than that in the other two groups (Figure 2E,F). In addition, absolute and relative soleus and plantaris muscle weight in the PHGG group were significantly higher than those in the ND and FFD groups (Figure 2G–J). Representative histology of soleus and plantaris muscle is shown in Figure 2K. The cross-sectional area of soleus and plantaris muscle in the PHGG group was significantly larger than that in the ND and FFD groups (Figure 2L,M).

### 3.6. Short-Chain Fatty Acids in Feces and Sera Increased by the Administration of PHGG

The concentration of short-chain fatty acids, such as acetate, propionate, and butyrate in feces and sera in the FFD group was lower than that in the ND group, whereas that in the PHGG groups was higher than that in the FFD group (Figure 3A–F).

### 3.7. The Administration of PHGG Significantly Increased Amino Acids Related to Muscle Protein Synthesis in the Plantaris Muscle, Whereas It Decreased Them in the Sera and Liver

Next, the concentration of amino acids in the plantaris muscle was measured. Amino acids related to muscle protein synthesis, such as valine, leucine, isoleucine, threonine, methionine, phenylalanine, and lysine, in the FFD group were significantly lower than that in the ND group, whereas those in the PHGG group were significantly higher than that in the FFD group (Figure 4A–G). On the other hand, the concentration of amino acids in the sera and liver of the PHGG group was lower than that of the ND group (Appendix A).

### 3.8. The Administration of PHGG Significantly Decreased Saturated Fatty Acid in the Plantaris Muscle

The concentration of palmitic acid, one of the saturated fatty acids, in the feces of the FFD group significantly decreased, compared to that of the ND group, whereas the palmitic acid concentration in the serum and plantaris muscle of the FFD group significantly increased. On the contrary, the palmitic acid concentration in the feces of the PHGG group significantly increased, compared to the ND and FFD groups, while the palmitic acid concentration in the serum and plantaris muscle of the PHGG group significantly decreased (Figure 4H–J).

### 3.9. The Administration of PHGG Decreased the Expression of Genes Related to Inflammation, Amino Acid Transporter, and Fatty Acid Transporter in the Small Intestine

In the RT-PCR analysis, the expression of genes related to inflammation in the jejunum was determined. The expression of genes related to inflammation, such as Tnfa, Il6, and Il1b in the jejunum of the FFD group, was significantly increased compared with that in the ND group, whereas those expressions decreased following administration of PHGG (Figure 5A–C). 

Similarly, the expression of genes related to fatty acid transporter, such as Cd36, and amino acid transporter, such as Slc15a, in the jejunum of the FFD group was significantly higher than that in the ND group, and the expression in the PHGG group was significantly lower than that in FFD group (Figure 5D,E). Moreover, the gene expression of Il-22 in the jejunum of the FFD group was significantly lower than that in the ND group, while that in the PHGG group was significantly higher than the expression level in the FFD group (Figure 5F).

A representative image of jejunum histology is shown in Appendix A. Villus height and width of the FFD groups were smaller than those of the ND group, while those of the PHGG groups were larger than those of the ND and PHGG group (Appendix A). Conversely, the crypt depth of the FFD group was larger than that of the ND and PHGG groups (Appendix A).

### 3.10. The Administration of PHGG Decreased the Expression of Genes Related to Muscle Atrophy and Inflammation

In the RT-PCR analysis, the expression of genes related to muscle atrophy and inflammation in the soleus muscle was determined. The expression of genes related to muscle atrophy, such as Trim63, Fbxo32, Foxo1, and Hdac4, in the FFD group, was significantly increased compared with in the ND group, whereas the expression decreased following administration of PHGG (Figure 5G–J). Similarly, the expression of genes related to inflammation, such as Tnfa and Il6, in the FFD group was significantly higher than that in the ND group, and expression in the PHGG group was significantly lower than that in the FFE group (Figure 5K,L).

### 3.11. The Administration of PHGG Regulated Inflammatory Responses in Innate Immunity in the LPL of the Small Intestine

Finally, we investigated the ratio of M1 (pro-inflammatory) and M2 (anti-inflammatory) macrophages in the LPL of the small intestine. The ratio of M1 macrophages in the FFD group was significantly increased, compared with that in the ND group. In contrast, administration of PHGG decreased the ratio of M1 macrophages (Figure 6A). Moreover, the ratio of M2 macrophages in the FFD group was significantly lower than that in the ND group, whereas that in the PHGG group was significantly increased (Figure 6B). The M1/M2 macrophage ratio in Db/Db mice was significantly increased, compared with that in the ND group, and administration of PHGG decreased the ratio (Figure 6C).

The ratio of ILC1s in CD45-positive cells of the FFD group was significantly higher than that in the ND group, and administration of PHGG significantly decreased the ratio of ILC1s (Figure 6D). On the other hand, the ratio of ILC3s in CD45-positive cells of the FFD group was lower than that in the ND group, and the administration of PHGG increased this (Figure 6E). The ratio of ex-ILC3s in CD45-positive cells of the FFD group was higher than that in the ND group, while administration of PHGG decreased this, similar to in the case of ILC1s (Figure 6F).

## 4. Discussion

The present study suggests that, compared with mice treated with ND containing cellulose, an insoluble dietary fiber, and mice treated with FFD, in which cellulose was replaced with cornstarch, administration of PHGG might alter the absorption of nutrients from the intestine by increasing the production of SCFA in the intestine and decreasing the number of inflammatory cells or increasing the number of anti-inflammatory cells in the LPL of the small intestine. This was considered to be mediated by increasing the production of SCFA in the intestine and decreasing the number of inflammatory cells or increasing the number of anti-inflammatory cells in the LPL of the small intestine. As a result, PHGG improved glucose, lipid metabolism, NAFLD, increased amino acids related with muscle synthesis in skeletal muscle, decreased saturated fatty acid in sera and muscle, and prevented sarcopenic obesity. With the improvement of sarcopenic obesity, a significant increase in energy metabolism was observed. On the other hand, the heaviest bodyweight in the PHGG group was due to the maintenance of muscle mass and the reduction in osmotic diuresis associated with improvement in glucose metabolism. The present study is the first to demonstrate the preventive effect of PHGG on sarcopenic obesity.

PHGG (Sunfiber^®^, Taiyo Kagaku Co., Ltd., Yokkaichi, Japan) is a soluble dietary fiber obtained by hydrolyzing guar gum, a highly viscous galactomannan derived from the guar bean, an edible plant in India and Pakistan, with microbial enzymes. Whereas guar gum is highly viscous, PHGG is a low-viscosity soluble dietary fiber due to its low molecular weight, making it an ideal addition to various kinds of food. There are many reports that PHGG increases the production of SCFA in the intestine [21,40,41], and in this study, administration of PHGG also increased the concentration of SCFA in the feces and sera. In a human study, Takahashi et al. [42]. reported that the effect of PHGG could be attributed to the degradation of fiber by bacteria, which promotes the growth of bifidobacteria and lactobacilli. In addition, it has also been reported that selective increased growth of bifidobacteria and lactobacillus can modify the gut microbiota and improve symptoms of inflammatory bowel disease, particularly relieving pain and abdominal swelling [43,44]. In summary, these findings suggest that PHGG supplementation may restore the physiological balance of the gut microbiota. In an animal study, Shimada et al. [45]. reported that the administration of PHGG increased plasma GLP-1 concentration GLP-1 is secreted from the intestines in response to food intake, and acts on the pancreas to promote insulin secretion and lower postprandial blood glucose [46]. Furthermore, GLP-1 has been shown to suppress appetite by acting on the hypothalamic paraventricular nucleus in the brain [47]. These findings suggest that the enhancement of GLP-1 secretion by PHGG may have prevented sarcopenia obesity through the improvement of impaired glucose tolerance.

The intestine has a very large contact surface with the environment, and the intestinal barrier plays a very important role in preventing foreign substances from entering the body. Among them, the mucus barrier is one of the first lines of protection for the digestive tract [48]. Disruption of the mucus barrier has been reported to alter innate immunity in the intestinal tract. In particular, disruption of the mucosal barrier alters the number of ILC3s, which are key regulators of inflammation and infection at the mucosal barrier [49]. IL-22 secreted from ILC3 enhances STAT3-dependent expression of antimicrobial peptides, contributing to the maintenance of intestinal epithelial barrier function [50,51,52]. Conversely, when ILC3 is lost, the expression of IL-22 is reduced and the level of antimicrobial peptides articulated by intestinal epithelial cells is decreased; ILC3 displays plasticity and the expression of transcription factors RORγt and T-bet alters the function of ILC3 [53]. When stimulated with cytokines such as Interleukin-18, this promotes the proliferation of ILC3s via NF-κB and production of interleukin-22. In a previous study, T-bet-positive ILC3s were shown to produce IFN-γ and inhibit the production of IL-17 and IL-22 [54]. Therefore, T-bet-positive ILC3s function similarly to ILC1. Recently, it has become clear that ILCs express receptors for SCFAs, such as G protein-coupled receptor (GPR) 41 (aka free fatty acid receptor (FFAR)3) and GPR43 (FFAR2), which play an important role in ILC proliferation. It has been reported that they stimulate the activation of transcriptional phosphatidylinositol-3 kinase, STAT3, STAT5, and mechanical target of rapamycin (mTOR) [55]. Hence, it was speculated that the ameliorative effect of PHGG on various metabolic disorders may be associated with the anti-inflammatory effect of innate immunity by increasing SCFA production in the intestine. In a future study, we would like to clarify the mechanism by which PHGG improves the function of ILC3 in the small intestine, which is impaired by a high-fat diet, based on the signal pathway of PI3K and STAT3.

It is known that the accumulation of lipids in muscle results in the inability of skeletal muscle glucose metabolism to respond adequately to insulin signaling [56,57,58]. We have previously reported that the expression of CD36, a long-chain fatty acid transporter, is reduced in association with the improvement of intestinal inflammation by the increase in SCFAs [29]. In the present study, the expression of CD36 in the small intestine was also significantly decreased in the PHGG group. As a result, the concentration of palmitic acid, a saturated fatty acid, was significantly increased in excretion in the feces, whereas its concentration in the sera and skeletal muscle was significantly decreased compared to the other two groups. Several studies have suggested that saturated and unsaturated fatty acids differentially regulate skeletal muscle mass and function. The administration of palmitic acid in C2C12 myotubes has been shown to decrease myotube diameter and suppress insulin signaling [59]. In addition, it has been reported that supplying palmitic acid to myocytes induces the expression of hypertrophy-promoting genes such as atrogin-1/MAFbx and simultaneously increases the nuclear localization of its transcriptional regulator, FoxO3 [60]. In addition, the reason for the increase in weight in the FFD group after the PHGG group was due to the increase in visceral fat mass. We have previously shown that excessive saturated fatty acids are absorbed into the body and stored in visceral fat, causing inflammation in visceral fat and various metabolic abnormalities [61]. In summary, the inhibition of saturated fatty acid absorption from the small intestine by PHGG may not only prevent sarcopenia by improving insulin resistance in skeletal muscle and suppressing the expression of muscle atrophy-related genes but also prevent various metabolic disorders by suppressing accumulation in visceral fat and improving encephalopathy in visceral fat.

Insulin signaling is also essential for amino acid transport and metabolism in skeletal muscle and plays a key role in the regulation of muscle protein synthesis, particularly through the activation of the mammalian rapamycin complex 1 pathway [62,63]. In this study, the concentration of amino acids absorbed from the small intestine and accumulated in the liver was lower in the PHGG group than that in the other two groups, while the concentration of amino acids in the skeletal muscle of the PHGG group was significantly higher than in that of the other two groups, and the expression of Slc15a in the small intestine was also significantly decreased. This result is similar to the report that amino acid absorption from the intestinal tract is significantly increased in obese mice compared to normal mice [64,65]. In contrast, the increase in amino acid concentration in the skeletal muscle of the PHGG group may be due to the following: the increase in SCFAs in the intestine of the PHGG group activates ILC3s in LPL and enhances the mucosal protective effect of IL-22 released by ILC3s, which reduces the influx of inflammatory substances such as LPS and intestinal bacteria into the body, alleviates systemic inflammation, and suppresses inflammation within the skeletal muscle. Inflammation in the skeletal muscle has been reported to increase muscle insulin resistance [66], suggesting that the reduction in muscle insulin resistance may have enhanced amino acid absorption into skeletal muscle [62,63], as described above. Furthermore, energy metabolism in the PHGG group was also higher than that in the other two groups, which suggested that the amount of activity was higher in the PHGG group. Since it has been reported that amino acid production in the skeletal muscle increases with increased activity [67], further amino acid production in the skeletal muscle could be promoted, preventing sarcopenia associated with inactivity and obesity.

## 5. Conclusions

In conclusion, the present study is the first to show the preventive effect of PHGG on sarcopenic obesity by altering the absorption of nutrients from the intestine. Changes in nutrient absorption might be involved by promoting an anti-inflammatory shift of innate immunity in the intestine accompanied by an increase in SCFA production by PHGG.

## Figures and Tables

**Figure 1 nutrients-14-01157-f001:**
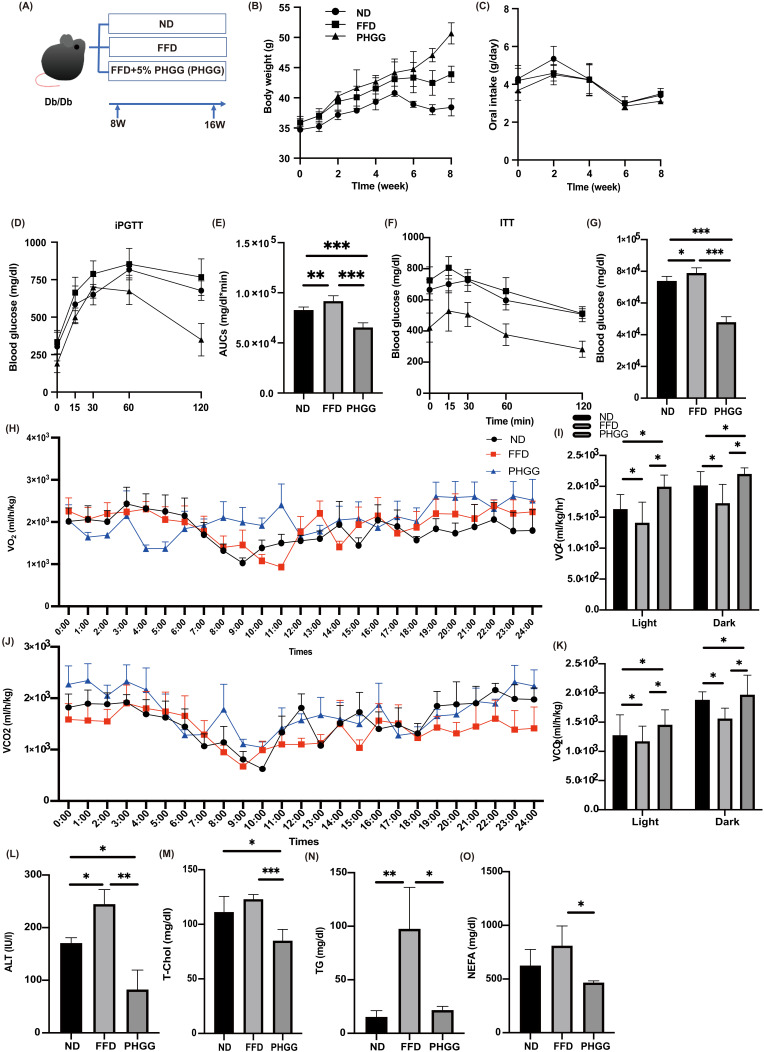
The administration of PHGG improved glucose tolerance, increased energy and metabolism, and improved hepatic enzymes and lipid metabolism. (**A**) Administration of PHGG (5% per feed weight) started at 8 weeks. (**B**) Changes in body weight (*n* = 6). (**C**) Changes in oral intake (*n* = 6). (**D**,**E**) Results of intraperitoneal glucose tolerance test (2 g/kg body weight) for 15-week-old mice and area under the curve (AUC) analysis (*n* = 6). (**F**,**G**) Results of insulin tolerance test (0.75 U/kg body weight) for 15-week-old mice and AUC analysis (*n* = 6). (**H**) Real-time monitoring curve of oxygen consumption (VO2) (*n* = 6). (**I**) Quantification of O2 consumption (*n* = 6). (**J**) Real-time monitoring curve of carbon dioxide release (VCO2) (*n* = 6). (**K**) Quantification of carbon dioxide release (*n* = 6). (**L**–**O**) Serum alanine aminotransferase (ALT), total cholesterol (T-Chol), triglyceride (TG), and non-esterified fatty acid (NEFA) levels (*n* = 6). Data are presented as mean ± SD. * *p* < 0.05, ** *p* < 0.01, *** *p* < 0.001, as determined by one-way ANOVA.

**Figure 2 nutrients-14-01157-f002:**
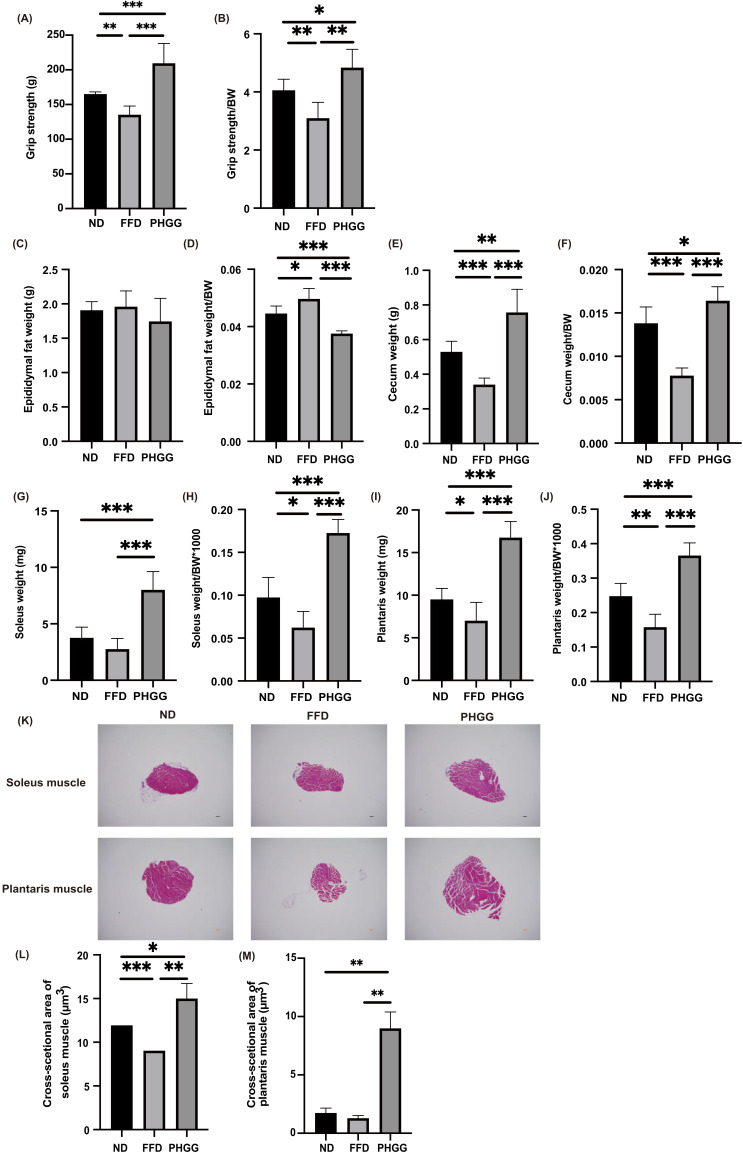
The administration of PHGG improved visceral fat obesity and sarcopenic obesity. (**A**,**B**) Absolute and relative grip strength in 16-week-old mice (*n* = 6 in each case). (**C**,**D**) Absolute and relative epididymal fat weight in 16-week-old mice (*n* = 6 in each case). (**E**,**F**) Absolute and relative cecum weight in 16-week-old mice (*n* = 6 in each case). (**G**,**H**) Absolute and relative soleus muscle weight, (**I**,**J**) absolute and relative plantaris muscle weight in 16 weeks old mice (*n* = 6 in each case). (**K**) Representative images of hematoxylin and eosin-stained soleus and plantaris muscle sections. Muscle tissues were collected at 16 weeks. The scale bar shows 100 μm. (**L**,**M**) The cross-sectional area of the soleus muscle (μm^3^) in 16-week-old mice (*n* = 6 in each case). Data are presented as mean ± SD; * *p <* 0.05, ** *p <* 0.01, *** *p <* 0.001, as determined by one-way ANOVA.

**Figure 3 nutrients-14-01157-f003:**
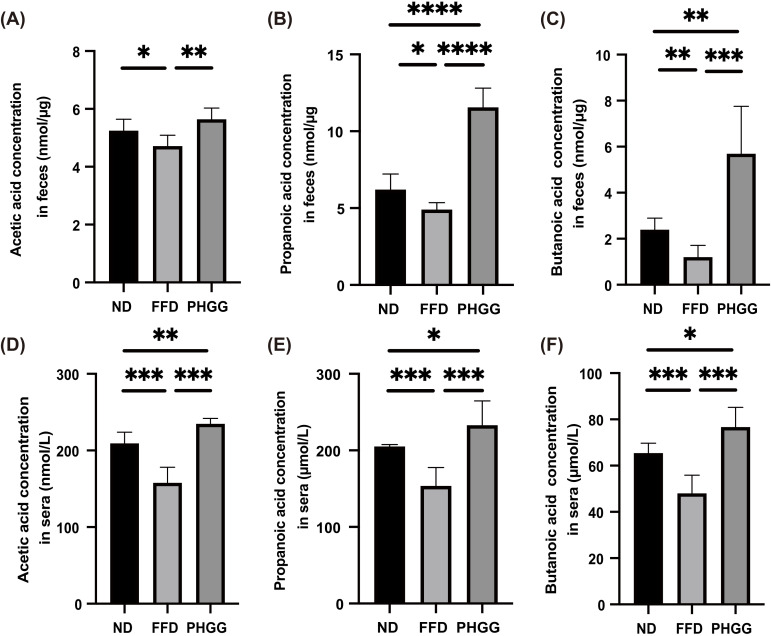
The administration of PHGG increased the concentration of short-chain fatty acid in feces and sera. The concentrations of (**A**) acetic acid, (**B**) propanoic acid, and (**C**) butanoic acid in feces (*n* = 6). The concentrations of (**D**) acetic acid, (**E**) propanoic acid, and (**F**) butanoic acid in sera (*n* = 6). Data are presented as mean ± SD; * *p <* 0.05, ** *p <* 0.01, *** *p <* 0.001, and **** *p <* 0.0001 as determined by one-way ANOVA.

**Figure 4 nutrients-14-01157-f004:**
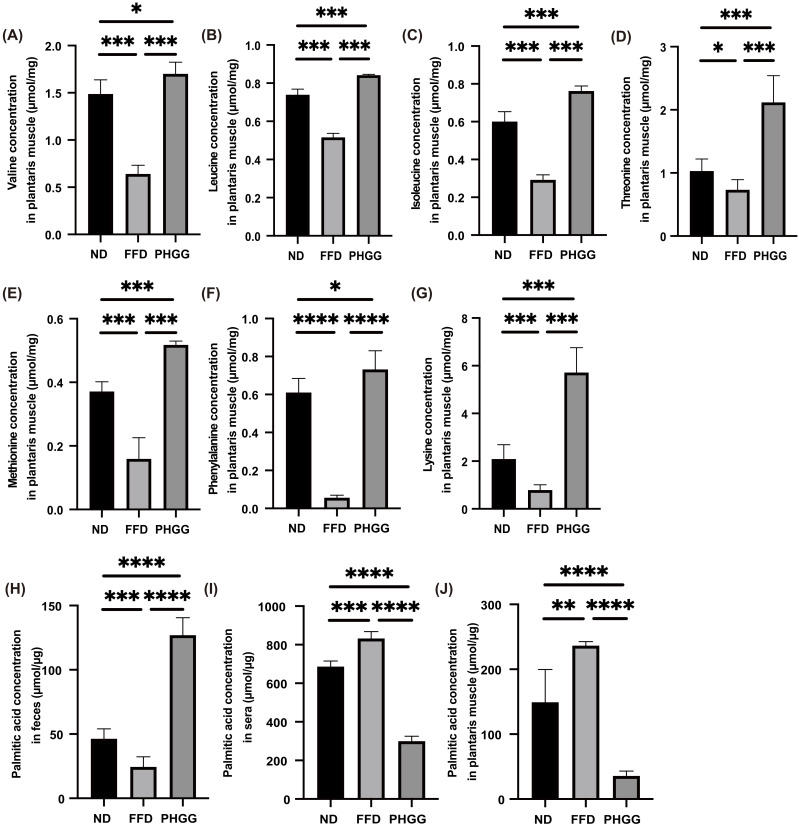
The administration of PHGG increased the concentration of amino acids related to muscle biosynthesis. The concentrations of (**A**) valine, (**B**) leucine, (**C**) isoleucine, (**D**) threonine, (**E**) methionine, (**F**) phenylalanine, and (**G**) lysine in the plantaris muscle (*n* = 6). The concentration of palmitic acid in (**H**) feces, (**I**) sera, and (**J**) plantaris muscle (*n* = 6). Data are presented as mean ± SD; * *p <* 0.05, ** *p <* 0.01, *** *p <* 0.001, and **** *p <* 0.0001 as determined by one-way ANOVA.

**Figure 5 nutrients-14-01157-f005:**
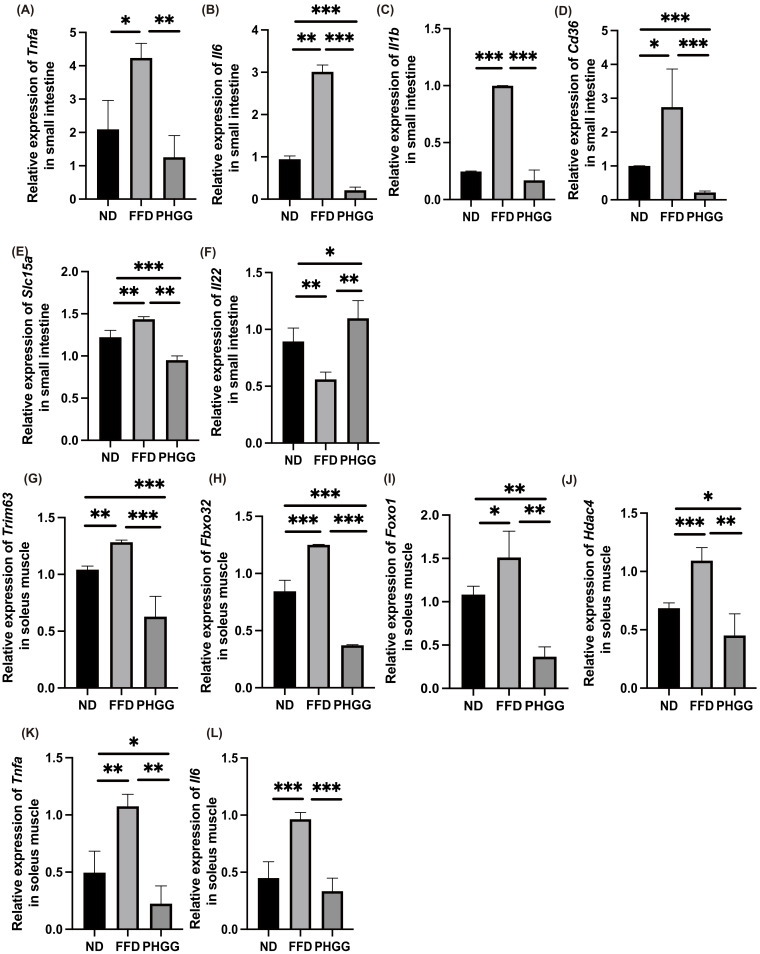
Administration of PHGG increased grip strength and muscle weight. Relative mRNA expression of (**A**) *Tnfa*, (**B**) *Il6*, (**C**) *Il1b*, (**D**) *Cd36*, (**E**) Slc15a, and (**F**) *Il22* mRNA expression in the small intestine normalized to the expression of GAPDH (*n* = 6). Relative mRNA expression of (**G**) *Trim63*, (**H**) *Fbxo32*, (**I**) *Foxo1*, (**J**) *Hdac4*, (**K**) *Tnfa*, and (**L**) *Il6* in the soleus muscle normalized to the expression of *GAPDH* (*n* = 6). Data are presented as mean ± SD. * *p <* 0.05, ** *p <* 0.01, *** *p <* 0.001, as determined by one-way ANOVA.

**Figure 6 nutrients-14-01157-f006:**
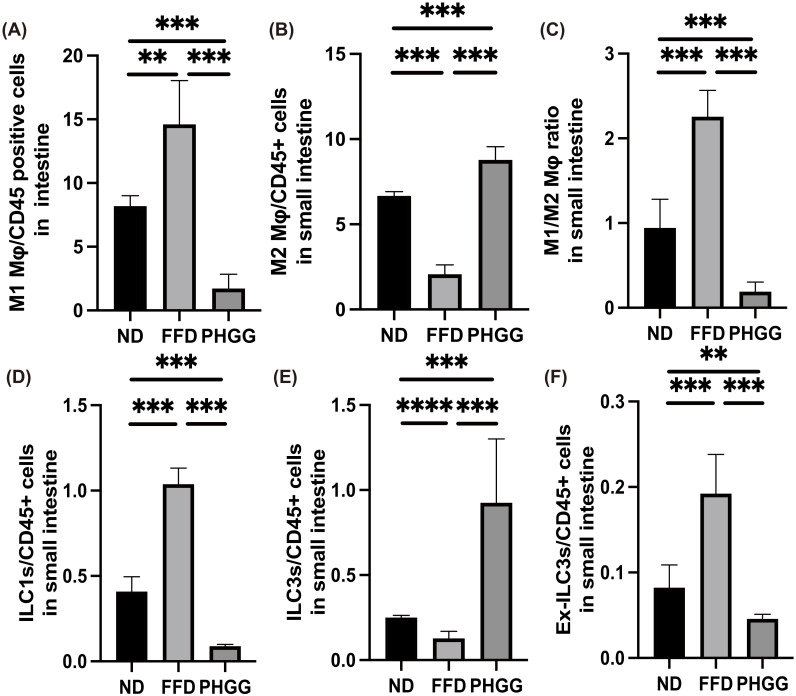
Administration of PHGG decreased the ratio of M1 macrophages, ILC1, and ILC3, and the ratio of M2 macrophages and ILC2. Ratio of (**A**) M1 macrophages to CD45-positive cells, (**B**) M2 macrophages to CD45-positive cells, (**C**) M1 to M2 macrophages, (**D**) ILC1s to CD45-positive cells, (**E**) ILC3s to CD45-positive cells, and (**F**) Ex-ILC3s CD45-positive cells in the small intestine (*n* = 6 in each case). Data are presented as mean ± SD. ** *p <* 0.01, *** *p <* 0.001, and **** *p <* 0.0001 as determined by one-way ANOVA.

**Table 1 nutrients-14-01157-t001:** Composition of normal diet, fiber-free diet, and diet with partially-hydrolyzed guar gum.

Ingredients (g/kg)	ND	FFD	PHGG
PHGG	0	0	50
Cellulose	50	0	0
Corn starch	397.486	447.486	397.486
Casein		200	
L-cystine		3	
Sucrose		100	
Soybean oil		70	
AIN-93 mineral mixture		35	
AIN-93G vitamin mixture		10	

FFD, fiber-free diet; ND, normal diet; PHGG, partially-hydrolyzed guar gum.

## Data Availability

Data available on request due to restrictions (privacy or ethical).

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
