# Peer review of "Partially Hydrolyzed Guar Gum Suppresses the Development of Sarcopenic Obesity"

_nutrients, 2022, doi:10.3390/nu14061157_

Round 1

Reviewer 1 Report

Overall, the research article "Partially hydrolyzed Guar Gum Suppresses the Development of Sarcopenic Obesity" is well written by Okamura, T. et al., and I have only minor suggestion like the Oil Red O Staining protocol is missing in the methods section, and the discussion could be improved much better.

There are some additional periods and spaces in some headings.  

Line 201 it should be read as 10% buffered formaldehyde fixed and embedded in paraffin,

line 296 it must be cecum instead of appendix,

line 408-414 is a complex sentence,

and in line 411 and 412 it must be....decrease the number of inflammatory cells or increase the number of anti-inflammatory cells...,

line 443 PI3K sentence is not complete,

line 482 could be read as due to the increase in SCFAs.  

Reference(s) are missing in lines 434 and 438, and in line 440 only one reference is mentioned for previous studies.

Author Response

  1. There are some additional periods and spaces in some headings.   Response Thank you for your kind comment. We have modified them.    
  2.  Line 201 it should be read as 10% buffered formaldehyde fixed and embedded in paraffin,   Response Thank you for your kind comment. As you say, we have modified the sentence described as below.   Materials and Methods “The small intestine tissue was obtained and fixed with 10% buffered formaldehyde and embedded in paraffin.”    
  3.  line 296 it must be cecum instead of appendix,   Response Thank you for your kind comment. As you say, we have modified the sentence described as below.   Results “Moreover, absolute and relative cecum weight in the FFD group was significantly lower than that in the ND group, whereas that in the PHGG group was significantly higher than that in the other two groups (Fig. 2 E and F).”   Figure legends “(E and F) Absolute and relative cecum weight in 16-weeks-old mice (n = 6 in each case).”    
  4.  line 408-414 is a complex sentence, and in line 411 and 412 it must be....decrease the number of inflammatory cells or increase the number of anti-inflammatory cells...,   Response Thank you for your kind comment. As you say, we have modified the sentences described as below.   Discussion “The present study suggests that, compared with mice treated with ND containing cellulose, an insoluble dietary fiber, and mice treated with FFD in which cellulose was replaced with cornstarch, administration of PHGG might alter the absorption of nutrients from the intestine by increasing the production of SCFA in the intestine and decrease the number of inflammatory cells or increase the number of anti-inflammatory cells in the LPL of the small intestine. As a result, PHGG improved glucose, lipid metabolism, NAFLD, increased of amino acids related with muscle synthesis in skeletal muscle, decreased of saturated fatty acid in sera and muscle, and prevented sarcopenic obesity.”    
  5.  line 443 PI3K sentence is not complete,   Response Thank you for your kind comment. According to your comment, we have modified the sentences described as below.     Discussion “Recently, ILCs express receptors for SCFAs, such as G protein-coupled receptor (GPR) 41 (also known as a free fatty acid receptor [FFAR]3) and GPR43 (FFAR2), which are important for ILC proliferation. It has been reported that they stimulate the activation of transcriptional phosphatidylinositol-3 kinase (PI3K), STAT3, STAT5, and mechanical target of rapamycin (mTOR) [49].”    
  6.  line 482 could be read as due to the increase in SCFAs.   Response Thank you for your kind comment. According to your comment, we have modified the sentence described as below.   Discussion “In contrast, the increase in amino acid concentration in the skeletal muscle of the PHGG group may be due to the following: the increase in SCFAs in the intestine of the PHGG group activates ILC3s in LPL and enhances the mucosal protective effect of IL-22 released by ILC3s, which reduces the influx of inflammatory substances such as LPS and intestinal bacteria into the body, alleviates systemic inflammation, and suppresses inflammation within skeletal muscle.”    
  7.  Reference(s) are missing in lines 434 and 438, and in line 440 only one reference is mentioned for previous studies.   Response Thank you for your valuable suggestion. When we re-read the relevant part, we found that sentence “IL-12 and IL-18 … to environmental cues” was not in the right context, so we have deleted it.

Reviewer 2 Report

This study entitled "Partially Hydrolyzed Guar Gum Suppresses the Development of Sarcopenic Obesity." was designed to preventive effect of PHGG on sarcopenic obesity. The authors found changes in nutrient absorption might be involved by promoting an anti-inflammatory shift of innate immunity in the intestine accompanied by an increase in SCFA production by PHGG. This study demonstrates a new concept and application of PHGG for sarcopenic obesity. However, there remains some important issues to be improved.

Major revision

  1. It seems more appropriate to explain sarcopenic obesity than to start by talking about type 2 diabetes in the introduction.
  2. It would be good to explain in detail the relationship between gut microbiota and sarcopenic obesity in the introduction.
  3. It is better to explain the hypothesis to be confirmed in this study in more detail in the introduction. In the present state, it is not clear why this study is necessary.
  4. Did you manufacture PHGG yourself? Or is it a commercial product? Please mention it in the method section.
  5. It is known that PI3K and STAT3 signaling play an important role in sarcopenic obesity. In this study, it was mentioned that it is related to PI3K and STAT3 signaling. Therefore, it would be good to check the expression of proteins involved in PI3K and STAT3 signaling.

Minor revision

  1. Check the unit notation. (ex. hours, hr -> h)
  2. Write the 2 of O2 and CO2 as a subscript.
  3. Remove the dot before the title as follow section. (2.1. 2.7. 3.5)

Author Response

Major revision

  1. It seems more appropriate to explain sarcopenic obesity than to start by talking about type 2 diabetes in the introduction. Response Thank you for your valuable suggestion. As you say, we focus on sarcopenic obesity in this study, therefore, to start by talking about type 2 diabetes is inappropriate. According to your comment, we have deleted the sentences in the Introduction described as below.   Introduction “The number of individuals with type 2 diabetes is rapidly increasing worldwide. Complications of type 2 diabetes reduce a person's quality of life, and they add a heavy burden to the medical economy [1]. The prevention of the progression of diabetic complications is thus an important task.”    
  2.  It would be good to explain in detail the relationship between gut microbiota and sarcopenic obesity in the introduction.   Response Thank you for your valuable suggestion. According to your comment, we have added the sentences in the Introduction described as below.   Introduction “Furthermore, dysbiosis has been reported to contribute to intestinal barrier dysfunction, which facilitates the transfer of lipopolysaccharide and endotoxin into the circulation, leading to systemic chronic inflammation and insulin resistance, and eventually to sarcopenia [a]. Since there are many reports on the relationship between obesity and dysbiosis, it is expected that there is also a relationship between sarcopenia obesity and altered gut microbiota.”   References a. Liao, X.; Wu, M.; Hao, Y.; Deng, H. Exploring the Preventive Effect and Mechanism of Senile Sarcopenia Based on “Gut-Muscle Axis.” Frontiers in bioengineering and biotechnology 2020, 8, doi:10.3389/FBIOE.2020.590869.  
  3.  It is better to explain the hypothesis to be confirmed in this study in more detail in the introduction. In the present state, it is not clear why this study is necessary.   Response Thank you for your valuable comment. In this study, we would like to emphasize that PHGG may prove to be an effective new treatment for sarcopenic obesity for which no effective treatment has ever been shown. Therefore, we have added the sentences described as below.   Introduction Muscle atrophy is also a risk factor for both decreased daily life activity and mortality [7,8], and, in particular, sarcopenic obesity has been reported to accelerate the loss of muscle mass and function, reduce physical performance, and increase the risk of death [a], but no effective treatment has been found.   “For the above reasons, we hypothesized that PHGG might be an effective new treatment for sarcopenia obesity, which has not been clarified.”    
  4.  Did you manufacture PHGG yourself? Or is it a commercial product? Please mention it in the method section.   Response Thank you for your valuable comment. The PHGG used in this study was provided by Taiyo Kagaku, and we asked Oriental Bio Service to mix it with AIN-93G to create a solid food.  Therefore, we have added the sentence described as below.   Materials and Methods “The PHGG used in this study was provided by Taiyo Kagaku, and we asked Oriental Bio Service to mix it with AIN-93G to create a solid food.”    
  5.  It is known that PI3K and STAT3 signaling play an important role in sarcopenic obesity. In this study, it was mentioned that it is related to PI3K and STAT3 signaling. Therefore, it would be good to check the expression of proteins involved in PI3K and STAT3 signaling.   Response Thank you for your valuable suggestion. As you say, to check the expression of proteins involved in PI3K and STAT3 signaling is important in this study. However, the Editor has instructed me to create a revise within 10 days, and we will not conduct this experiment in this study, but will pursue it in the next study. Therefore, we have added the following sentence to Discussion section.   Discussion In future study, we would like to clarify the mechanism by which PHGG improves the function of ILC3 in the small intestine, which is impaired by a high-fat diet, based on the signal pathway of PI3K and STAT3.  

Minor revision

  1. Check the unit notation. (ex. hours, hr -> h) Response Thank you for your kind comment. The notation has been unified to hours.
  2.  Write the 2 of O2 and CO2 as a subscript.   Response   Thank you for your kind comment. We have modified them.    
  3.  Remove the dot before the title as follow section. (2.1. 2.7. 3.5) Response Thank you for your kind comment. We have modified them.

Reviewer 3 Report

The English needs editing throughout. It is understandable, but there are numerous errors.

lines 33-34 - please combine these two sentences

lines 59-62- please add 'thus' or 'therefore prior to 'and PHGG could'

lines 62-64 - please rephrase. the sentence is not understandable

line 63- please remove 'therefore'

line 65 - it is unclear to what 'hypothetical mechanism' does the author refer to.  

line 72 - where does the fat come from in ND (AIN93)?

lines 73-75 - I the following sentence 'or modified 74 AIN93G rodent diet replaced cellulose with commercially-available, partially-hydrolyzed 75 guar gum' I would suggest to replace the word 'replaced' with 'in which the cellulose was replaced with...'

lines 80-81- please avoid using the word 'killed'

line 81 - why did the authors decide to use a mixture of anesthetic agents over ketamine or cervical dislocation which is widely used worldwide? Was the anesthetic injected ip - this was not mentioned.

lines 68-82 - in this paragraph the author did not mention anything about how the animals were housed, what were the humidity%, light cycle, cage size etc. please provide. How often was the feed replaced and how did you measure the intake (was it a manual measurement?-please clarify)?

Was the fresh chow provided to each group (ND, FFD, PHGG) every day? If so, was it done in the morning, evening (please add the exact time or a timeframe). Were the remnants of chow reused?

Figure 1B and 1C - time on X axis starts at week 8. I would suggest to start counting from week 0 as a baseline. Has the investigator noticed any difference in feces consistency between groups?

Figure 1C - how would the authors explain the drop in food intake between week 12 and 14 (effect observed in all groups)? 

I would suggest to create a table that lists all Ingredients and the composition of each experimental diet

lines 83-90 - please provide a reference to the method

line 88 - where was the blood taken from?

lines 105-106 - please rephrase to include that investigators were blinded to the experimental conditions

lines 108-112 -please provide a reference to the method

line 130 - what method was used to perform the biochemical examination. Please describe or provide a reference 

line 136  - Were the segments from the distal colon of each animal resected to get the fecal pellets or were the fecal pellets collected from the cage once expelled? In case the second method was used is it possible that the time of fecal collection could have impacted the GC-MS fatty acid analysis (different consistency of fecal pallets i.e. some could have died out if left in the cage for too long).

line 149 - GC-MS abbreviation was already explained earlier in the text. Authors can use GCMS 

line 187 - please consider changing Tnfa to either TNF-alpha or TNFα

line 230- please add a short paragraph describing how the result analysis was performed and what tests (including any post-hoc tests) were used.

line 245 - Why there are no results for VO2 between 0:00 and 4:00 (figure 1H) for PHGG?

line 271-273 - please rephrase as its difficult to understand. I would also avoid using 'gene expression related to inflammation, such as TNF' . I would suggest instead 'the expression of genes involved in the inflammatory response

lines 408-415- please rephrase this sentence as its difficult to understand. Please create several shorter sentences.

line 421- please consider changing 'derived by' to 'obtained by'

lines 497-498 - please rephrase for clarity

line 427 - what could be a potential side effect of PHGG intake for humans. Are there any reports showing its effect on gastrointestinal motility ? are there any reports showing the effect of PHGG on GLP-1? 

line 428 - please consider removing 'furthermore'. What 'disruption of the mucus barrier' the author is referring to?

lines 499-500- the following phrase 'further clinical studies' implies that there have already been clinical trials done - if thats the case please provide  the clinical trial gov identifier # and give more details of what exactly was measured and what were the main objectives of trials

Author Response

  1. lines 33-34 - please combine these two sentences. Response Thank you for your valuable suggestion. According to the other reviewer’s comment, we have deleted the sentences.  
  2.  lines 59-62- please add 'thus' or 'therefore prior to 'and PHGG could'. Response Thank you for your valuable suggestion. According to your comment, we have modified the sentence described as below. Introduction “Thus, PHGG alters the composition of the microbiota, and increases metabolites such as butyric acid, acetic acid, and various amino acids [24–26], therefore, PHGG could have beneficial health effects on the host through changes in the gut microbiota and its metabolites.”  
  3.  lines 62-64 - please rephrase. the sentence is not understandable. Response Thank you for your valuable suggestion. According to your comment, we have modified the sentences and added the references described as below. Introduction “Furthermore, dysbiosis has been reported to contribute to intestinal barrier dysfunction, which facilitates the transfer of lipopolysaccharide and endotoxin into the circulation, leading to systemic chronic inflammation and insulin resistance, and eventually to sarcopenia [a]. Since there are many reports on the relationship between obesity and dysbiosis, it is expected that there is also a relationship between sarcopenia obesity and altered gut microbiota. In addition, PHGG supports the effects of voluntary exercise on obesity and glucose intolerance due to a high-fat diet [b]. For the above reasons, we hypothesized that PHGG might be an effective new treatment for sarcopenia obesity, which has not been clarified.” References a. Liao, X.; Wu, M.; Hao, Y.; Deng, H. Exploring the Preventive Effect and Mechanism of Senile Sarcopenia Based on “Gut-Muscle Axis.” Frontiers in bioengineering and biotechnology 2020, 8, doi:10.3389/FBIOE.2020.590869. b. Aoki, T.; Oyanagi, E.; Watanabe, C.; Kobiki, N.; Miura, S.; Yokogawa, Y.; Kitamura, H.; Teramoto, F.; Kremenik, M.J.; Yano, H. The Effect of Voluntary Exercise on Gut Microbiota in Partially Hydrolyzed Guar Gum Intake Mice under High-Fat Diet Feeding. Nutrients 2020, 12, 1–13, doi:10.3390/nu12092508.  
  4.  line 63- please remove 'therefore'. Response Thank you for your valuable suggestion. According to your comment, we have modified the sentence described as below. Introduction “The present study was designed to examine the influence of PHGG on sarcopenic obesity and to investigate the related hypothetical mechanism.”  
  5.  line 65 - it is unclear to what 'hypothetical mechanism' does the author refer to.  Response Thank you for your valuable comment. According to your comment, we have added the sentence and reference described in comment 3.  
  6.  line 72 - where does the fat come from in ND (AIN93)? Response Thank you for your valuable comment. According to your comment, we have modified the sentence described as below.   Materials and Methods “They were fed for eight weeks either a normal diet (ND; AIN-93G, 361.2 kcal/100 g, fat kcal 7.0 % soybean oil, cellulose 5%/w; CLEA Japan, Tokyo),”  
  7.  lines 73-75 - the following sentence 'or modified ' I would suggest to replace the word 'replaced' with 'in which the cellulose was replaced with...' Response Thank you for your valuable comment. According to your comment, we have modified the sentence described as below.   Materials and Methods “a fiber-free diet (FFD; Modified AIN93G rodent diet in which the cellulose was replaced with corn starch [5%/w], 388.1 kcal/100 g), or modified AIN93G rodent diet in which the cellulose was replaced with commercially-available,”  
  8.  lines 80-81- please avoid using the word 'killed' Response Thank you for your valuable comment. According to your comment, we have modified the sentence described as below.   Materials and Methods When they were 16 weeks old, after an overnight fast, all the mice were sacrificed by the administration of a combination anesthetic: 0.3 mg/kg of medetomidine, 4.0 mg/kg, midazolam, and 5.0 mg/kg of butorphanol [29].  
  9.  line 81 - why did the authors decide to use a mixture of anesthetic agents over ketamine or cervical dislocation which is widely used worldwide? Was the anesthetic injected ip - this was not mentioned. Response Thank you for your valuable comment. Since ketamine was designated as a narcotic in 2007 in Japan, the combination anesthetic has been used in recent years for experimental animals as a narcotic-free alternative to the ketamine/xylazine mixture. In addition, we do not employ cervical dislocation when sacrificing mice because many researchers in our laboratory are not skilled in mouse dissection. Therefore, we have added the sentence described as below.   Materials and Methods “Since ketamine was designated as a narcotic in 2007 in Japan, the combination anesthetic has been used in our laboratory.”  
  10.  lines 68-82 - in this paragraph the author did not mention anything about how the animals were housed, what were the humidity%, light cycle, cage size etc. please provide. How often was the feed replaced and how did you measure the intake (was it a manual measurement?-please clarify)? Was the fresh chow provided to each group (ND, FFD, PHGG) every day? If so, was it done in the morning, evening (please add the exact time or a timeframe). Were the remnants of chow reused? Response Thank you for your valuable comment. According to your comment, we have added the sentences described as below.   Materials and Methods The mice were maintained in an environmentally controlled room (temperature, 23 ± 1.5°C; humidity, 40-60%) with a 12-h light/12-h dark cycle (7 a.m. to 7 p.m.). - 7 p.m.). Cages with the size of W220 × L320 × H135 mm were used, and three mice were kept in each cage. The cumulative oral intake was measured for 8 weeks. The manually weighed fresh food was placed in a trough in each cage once every three days in the morning 9 a.m., and after 24 h, the amounts of food were measured. The remnants of chow were discarded.  
  11.  Figure 1B and 1C - time on X axis starts at week 8. I would suggest to start counting from week 0 as a baseline. Has the investigator noticed any difference in feces consistency between groups? Response Thank you for your valuable suggestion. As you say we have modified the time of X axis in Figure 1B and 1C. Moreover, feces consistency in the PHGG group was slightly softer than that in ND and FFD group, however, feces consistency in the disease activity index of inflammatory bowel disease of the three groups was all formed.  
  12.  Figure 1C - how would the authors explain the drop in food intake between week 12 and 14 (effect observed in all groups)? Response Thank you for your valuable comment. We have used Db/Db mice in addition to the present study and experienced that most of the Db/Db mice tend to decrease their food intake after peaking at 10-12 weeks of age. We believe that the change in food intake between 14 and 16 weeks of age is a measurement error.  
  13.  I would suggest to create a table that lists all Ingredients and the composition of each experimental diet. Response Thank you for your valuable suggestion. According to your comment, we have added the table of all ingredients and the composition of each experimental diet.   Materials and Methods They were fed for eight weeks either a normal diet (ND; AIN-93G, 361.2 kcal/100 g, fat kcal 7.0 % soybean oil, cellulose 5%/w; CLEA Japan, Tokyo), a fiber-free diet (FFD; Modified AIN93G rodent diet in which the cellulose was replaced with corn starch [5%/w], 388.1 kcal/100 g), or modified AIN93G rodent diet in which the cellulose was replaced with commercially-available, partially-hydrolyzed guar gum (PHGG; Modified AIN93G rodent diet without cellulose added PHGG [5%/w], 378.2 kcal/100 g) (Sunfiber®, Taiyo Kagaku Co., Ltd., Mie, Japan) (Table 1).  
  14.  lines 83-90 - please provide a reference to the method. Response Thank you for your valuable suggestion. According to your comment, we have added a reference described as below.   References a. Kawano, R.; Okamura, T.; Hashimoto, Y.; Majima, S.; Senmaru, T.; Ushigome, E.; Asano, M.; Yamazaki, M.; Takakuwa, H.; Sasano, R.; et al. Erythritol Ameliorates Small Intestinal Inflammation Induced by High-Fat Diets and Improves Glucose Tolerance. International Journal of Molecular Sciences 2021, 22, doi:10.3390/ijms22115558.  
  15.  line 88 - where was the blood taken from? Response Thank you for your valuable comment. Blood was collected by orbital puncture. Therefore, we have modified the sentences described as below.   Materials and Methods “To measure weight, the mice were fasted for 14 hours overnight and the weight was measured once a week. In 15-week-old mice, an intraperitoneal glucose tolerance test (iPGTT) (2 g/kg body weight) and an insulin tolerance test (ITT) (0.5 U/kg body weight) were performed after 14-hours and 5-hours fasts, respectively, and blood glucose was measured with blood collected by orbital puncture and as indicated by a glucometer (Gultest Neo Alpha; Sanwa Kagaku Kenkyusho, Nagoya, Japan). The area under the curve (AUC) of the iPGTT and ITT results was analyzed [30].”  
  16.  lines 105-106 - please rephrase to include that investigators were blinded to the experimental conditions. Response Thank you for your kind comment. According to your comment, we have modified the sentence described as below.   Materials and Methods “and the investigators were blinded to the experimental conditions.”  
  17.  lines 108-112 -please provide a reference to the method Response Thank you for your valuable comment. According to your comment, we have added a reference described as below.   References a. Okamura, T.; Hashimoto, Y.; Osaka, T.; Senmaru, T.; Fukuda, T.; Hamaguchi, M.; Fukui, M. MiR-23b-3p Acts as a Counter-Response against Skeletal Muscle Atrophy. Journal of Endocrinology 2020, 244, 535–547, doi:10.1530/JOE-19-0425.  
  18.  line 130 - what method was used to perform the biochemical examination. Please describe or provide a reference  Response Thank you for your valuable suggestion. According to your comment, we have added the sentences and reference described as below.   Materials and Methods Blood samples were taken from fasted mice and the serum samples were collected after centrifugation at 14,000 rpm for 10 minutes at 4°C. The levels of alanine aminotransferase (ALT) were measured by standardization support method of the Japanese Society for Clinical Chemistry [a], and the levels of total cholesterol (T-Chol) [b], triglycerides (TG) [c], and non-esterified fatty acids (NEFAs) [d] were measured by enzymatic method . Biochemical examinations were performed using FUJIFILM Wako Pure 18 Chemical Corporation (Osaka, Japan).   References a. Kotani K, Maekawa M, Kanno T. [Reestimation of aspartate aminotransferase (AST)/alanine aminotransferase (ALT) ratio based on JSCC consensus method--changes of criteria for a differential diagnosis of hepatic disorders following the alteration from Karmen method to JSCC method]. Nihon Shokakibyo Gakkai Zasshi. 1994;91:154-161, in Japanese. b. Allain CC, Poon LS, Chan CS, Richmond W, Fu PC. Enzymatic determination of total serum cholesterol. Clin Chem. 1974;20:470-475. c. McGowan MW, Artiss JD, Strandbergh DR, Zak B. A peroxidase-coupled method for the colorimetric determination of serum triglycerides. Clin Chem. 1983;29:538-542. d. Christmass, M.A.; Mitoulas, L.R.; Hartmann, P.E.; Arthur, P.G. A Semiautomated Enzymatic Method for Determination of Nonesterified Fatty Acid Concentration in Milk and Plasma. Lipids 1998, 33, 1043–1049, doi:10.1007/S11745-998-0304-9.  
  19.  line 136  - Were the segments from the distal colon of each animal resected to get the fecal pellets or were the fecal pellets collected from the cage once expelled? In case the second method was used is it possible that the time of fecal collection could have impacted the GC-MS fatty acid analysis (different consistency of fecal pallets i.e. some could have died out if left in the cage for too long). Response Thank you for your valuable comment. Fecal pellets were collected by resection of the distal colon during sacrifice. Therefore, we have added the sentence described as below.   Materials and Methods “Fecal pellets were collected by resection of the distal colon during sacrifice.”  
  20.  line 149 - GC-MS abbreviation was already explained earlier in the text. Authors can use GCMS  Response Thank you for your kind comment. We have modified it.  
  21.  line 187 - please consider changing Tnfa to either TNF-alpha or TNFα Response Thank you for your kind comment. Since it represents a mouse gene, it is expressed in lowercase and italics.  
  22.  line 230- please add a short paragraph describing how the result analysis was performed and what tests (including any post-hoc tests) were used.   Response Thank you for your kind comment. As you say, statistical analyses section missed. Therefore, we have added the section of statistical analyses described as below.   Materials and Methods 2.14. statistical analysis “The data were analyzed using the JMP ver. 13.0 software (SAS, Cary, NC, USA). One-way analysis of variance was used to compare different groups. P-values <0.05 were accepted as statistically significant. Figures were generated using the GraphPad Prism software (version 9.0; San Diego, CA, USA).”  
  23.  line 245 - Why there are no results for VO2 between 0:00 and 4:00 (figure 1H) for PHGG? Response Thank you for your kind comment. We have modified it.  
  24.  line 271-273 - please rephrase as its difficult to understand. I would also avoid using 'gene expression related to inflammation, such as TNF' . I would suggest instead 'the expression of genes involved in the inflammatory response Response Thank you for your kind comment. According to your comment, we have modified the sentences described as below.   Results “In RT-PCR analyses, the expression of genes involved in the inflammatory response, such as Tnfa, Il6, and Il1b, and fibrosis, such as Col1a, in the liver of the FFD group was significantly higher than that of the ND group. On the other hand, that of the PHGG group was significantly lower than that of FFD group (Sup Fig. 1 D-G).”  
  25.  lines 408-415- please rephrase this sentence as its difficult to understand. Please create several shorter sentences. Response Thank you for your kind comment. According to your comment, we have modified the sentences described as below.   Discussion “The present study suggests that, compared with mice treated with ND containing cellulose, an insoluble dietary fiber, and mice treated with FFD in which cellulose was replaced with cornstarch, administration of PHGG might alter the absorption of nutrients from the intestine. This was considered to be mediated by increasing production of SCFA in the intestine and decrease in the number of inflammatory cells or increase in the number of anti-inflammatory cells in the LPL of the small intestine. As a result, PHGG improved glucose, lipid metabolism, NAFLD, increased of amino acids related with muscle synthesis in skeletal muscle, decreased of saturated fatty acid in sera and muscle, and prevented sarcopenic obesity.”  
  26.  line 421- please consider changing 'derived by' to 'obtained by' Response Thank you for your kind comment. As you say, we have modified it.   Discussion “PHGG (Sunfiber®, Taiyo Kagaku Co., Ltd., Mie, Japan) is a soluble dietary fiber obtained by hydrolyzing guar gum,”  
  27.  lines 497-498 - please rephrase for clarity Response Thank you for your valuable suggestion. According to your comment, we have modified the sentence described as below.   Conclusions “In conclusion, the present study is the first to show the preventive effect of PHGG on sarcopenic obesity by altering the absorption of nutrients from the intestine.”  
  28.  line 427 - what could be a potential side effect of PHGG intake for humans. Are there any reports showing its effect on gastrointestinal motility ? are there any reports showing the effect of PHGG on GLP-1?  Response Thank you for your valuable question. Thank you for your valuable question.  First, if you take too much PHGG at once, you may experience abdominal swelling, diarrhea, and nausea, according to the PHGG product description. In addition, there were no obvious side effects reported in an RCT of IBS patient (Niv E, Halak A, Tiommny E, Yanai H, Strul H, Naftali T, Vaisman N. Randomized clinical study: Partially hydrolyzed guar gum (PHGG) versus placebo in the treatment of patients with irritable bowel syndrome. Nutr Metab (Lond). 2016;13:10. doi: 10.1186/s12986-016-0070-5.). Therefore, it is highly unlikely that PHGG will cause any side effects if taken in proper dosage. Next, in several human studies, it has been suggested that the effects of PHGG may be attributed to the degradation of fiber by bacteria, which promotes the growth of bifidobacteria and lactobacilli.  In addition, it has also been reported that selective increased growth of bifidobacteria and lactobacillus can modify the gut microbiota and improve the symptoms of inflammatory bowel disease, particularly relieving pain and abdominal swelling. In summary, these findings may suggest that PHGG supplementation may restore the physiological balance of the gut microbiota. Finally, in an animal study using rats, PHGG and GLP-1 were reported to increase plasma GLP-1 levels in rats treated with PHGG. According to your comment, we have added the sentences and references described as below. Discussion “In a human study, Takahashi, et al [a]. reported that the effect of PHGG could be attributed to the degradation of fiber by bacteria, which promotes the growth of bifidobacteria and lactobacilli. In addition, it has also been reported that selective increased growth of bifidobacteria and lactobacillus can modify the gut microbiota and improve the symptoms of inflammatory bowel disease, particularly relieving pain and abdominal swelling [b,c]. In summary, these findings may suggest that PHGG supplementation may restore the physiological balance of the gut microbiota. In an animal study, Shimada, et al [d]. reported that the administration of PHGG increased the plasma GLP-1 concentration in plasma. GLP-1 is secreted from the intestines in response to food intake, and acts on the pancreas to promote insulin secretion and lower postprandial blood glucose [e]. Furthermore, GLP-1 has been shown to suppress appetite by acting on the hypothalamic paraventricular nucleus in the brain [f]. These findings suggest that the enhancement of GLP-1 secretion by PHGG may have prevented sarcopenia obesity through the improvement of impaired glucose tolerance.”   References a. Yamamoto, T.; Wako, N.; Okubo, T.; Ishihara, N. Influence of Partially Hydrolyzed Guar Gum on Constipation in Women. Journal of nutritional science and vitaminology 1994, 40, 251–259, doi:10.3177/JNSV.40.251. b. Nobaek, S.; Johansson, M.-L.; Molin, G.; Ahrné, S.; Jeppsson, B. Alteration of Intestinal Microflora Is Associated with Reduction in Abdominal Bloating and Pain in Patients with Irritable Bowel Syndrome. The American journal of gastroenterology 2000, 95, 1231–1238, doi:10.1111/J.1572-0241.2000.02015.X. c. Saggioro, A. Probiotics in the Treatment of Irritable Bowel Syndrome. Journal of clinical gastroenterology 2004, 38, doi:10.1097/01.MCG.0000129271.98814.E2. d. Shimada, R.; Yoshimura, M.; Murakami, K.; Ebihara, K. Plasma Concentrations of GLP-1 and PYY in Rats Fed Dietary Fiber Depend on the Fermentability of Dietary Fiber and Respond to an Altered Diet. International Journal of Clinical Nutrition & Dietetics 2015, 2015, doi:10.15344/2456-8171/2015/103. e. MacDonald, P.E.; El-kholy, W.; Riedel, M.J.; Salapatek, A.M.F.; Light, P.E.; Wheeler, M.B. The Multiple Actions of GLP-1 on the Process of Glucose-Stimulated Insulin Secretion. Diabetes 2002, 51 Suppl 3, doi:10.2337/DIABETES.51.2007.S434. f. Paone, P.; Cani, P.D. Mucus Barrier, Mucins and Gut Microbiota: The Expected Slimy Partners? Gut 2020, 69, 2232–2243, doi:10.1136/GUTJNL-2020-322260.  
  29.  line 428 - please consider removing 'furthermore'. What 'disruption of the mucus barrier' the author is referring to? Response Thank you for your valuable comment. According to you comment, we have removed the word. And, disruption of the mucus barrier meant thinning of the mucus barrier, one of the intestinal barriers that protects the digestive tract. As you say, the following sentences and reference were added to the front of this sentence, as it requires an explanation of the mucus barrier.   Discussion “The intestine has a very large contact surface with the environment, and the intestinal barrier plays a very important role in preventing foreign substances from entering the body. Among them, the mucus barrier is one of the first lines of protection for the digestive tract [a]. Disruption of the mucus barrier has been reported to alter innate immunity in the intestinal tract.”   References a. Paone, P.; Cani, P.D. Mucus Barrier, Mucins and Gut Microbiota: The Expected Slimy Partners? Gut 2020, 69, 2232–2243, doi:10.1136/GUTJNL-2020-322260.  
  30.  lines 499-500- the following phrase 'further clinical studies' implies that there have already been clinical trials done - if that the case please provide the clinical trial gov identifier # and give more details of what exactly was measured and what were the main objectives of trials Response Thank you for your valuable suggestion. In fact, the research plan has been prepared and is currently under review by the ethical committee of our university. Therefore, we have deleted the sentence.